# Documentation of Psychosocial Distress and Its Antecedents in Children with Rare or Life-Limiting Chronic Conditions

**DOI:** 10.3390/children9050664

**Published:** 2022-05-05

**Authors:** Sarah R. McCarthy, Elizabeth H. Golembiewski, Derek L. Gravholt, Jennifer E. Clark, Jeannie Clark, Caree Fischer, Hannah Mulholland, Kristina Babcock, Victor M. Montori, Amie Jones

**Affiliations:** 1Department of Psychiatry and Psychology, Mayo Clinic, Rochester, MN 55905, USA; 2Knowledge and Evaluation Research Unit, Mayo Clinic, Rochester, MN 55905, USA; golembiewski.elizabeth@mayo.edu (E.H.G.); gravholt.derek@mayo.edu (D.L.G.); montori.victor@mayo.edu (V.M.M.); 3Department of Endocrinology, Diabetes, and Metabolism, Mayo Clinic College of Medicine and Science, Mayo Clinic, Rochester, MN 55905, USA; clark.jennifer1@mayo.edu; 4Mayo Clinic Children’s Center, Mayo Clinic, Rochester, MN 55905, USA; clark.jeannie@mayo.edu (J.C.); fischer.caree@mayo.edu (C.F.); 5Section of Social Work, Mayo Clinic, Rochester, MN 55905, USA; mulholland.hannah@mayo.edu (H.M.); donahue.kristina@mayo.edu (K.B.); 6Department of Pediatric and Adolescent Medicine, Mayo Clinic, Rochester, MN 55905, USA; jones.amie@mayo.edu

**Keywords:** psychosocial distress, pediatrics, complex chronic conditions, rare diseases

## Abstract

Children with rare or life-limiting chronic conditions and their families are at high risk of psychosocial distress. However, despite its impact on patient and family health and functioning, psychosocial distress and its antecedents may not routinely be captured in medical records. The purpose of this study was to characterize current medical record documentation practices around psychosocial distress among children with rare or life-limiting chronic conditions and their families. Medical records for patients with rare or life-limiting chronic conditions (*n* = 60) followed by a pediatric complex care program were reviewed. Study team members extracted both structured data elements (e.g., diagnoses, demographic information) and note narratives from the most recent visit with a clinician in the program. Psychosocial topics were analyzed using a mixed quantitative (i.e., frequency counts of topics) and qualitative approach. Topics related to psychosocial distress that were documented in notes included child and parent emotional problems, parent social support, sibling emotional or physical problems, family structure (e.g., whether parents were together), and financial concerns. However, 35% of notes lacked any mention of psychosocial concerns. Although examples of psychosocial concerns were included in some notes, none were present in over one-third of this sample. For both patients with rare or life-limiting chronic conditions and their caregivers, more active elicitation and standard documentation of psychosocial concerns may improve the ability of healthcare providers to identify and intervene on psychosocial concerns and their risk factors.

## 1. Introduction

Over the past 50 years, medical advances have reduced overall childhood morbidity and mortality, allowing children with rare or life-limiting chronic conditions to survive and live longer [1,2,3,4,5]. Rare or life-limiting chronic conditions include congenital abnormalities, neurodegenerative diseases, metabolic disorders, the sequelae of extreme prematurity, and other disorders that are often serious and incurable but may be managed with ongoing medical intervention and lifestyle adaptations. However, despite medical and scientific advances, children with rare or life-limiting chronic conditions and their families often face significant challenges to their quality of life as a result of long-term, complex medical regimens [6], frequent provider visits and hospitalizations [3], gaps in care coordination [7,8], and functional limitations that are typically severe and may include reliance on technology [8,9,10]. In addition to healthcare and functional challenges, children with rare or life-limiting chronic conditions are at increased risk for the development of social [11], behavioral [12], and emotional problems [13], which, if not detected and treated, can impact the child’s adherence to medical recommendations [14,15], exacerbate physical illness [16], and increase healthcare utilization [17].

The struggles associated with childhood rare or life-limiting chronic conditions are not limited to the patients themselves, as parents and other family caregivers assume tremendous responsibility on behalf of these vulnerable patients [18]. Collectively, caregivers for children with rare or life-limiting chronic conditions have been described as a “shadow” healthcare system for children with medical complexity [19], forced to act as patient advocates, care coordinators, and home health aides, resulting in significant disruptions to work and family function [20,21,22]. Not surprisingly, parents of children with rare or life-limiting chronic conditions frequently report problems related to mood [23], physical function [24], marital discord [25], social isolation [22], and unmet needs [26]. Many parents of children with rare or life-limiting chronic conditions experience disruption to their careers [27,28] and financial insecurity [29]. High levels of parental distress, in turn, can impact a child’s medication adherence [30] and has been linked to greater emotional distress and reduced quality of life in the child [31,32]; highlighting the importance of attention to parental emotional functioning in the context of the child’s care [33].

Together, these behavioral, emotional, social, and financial challenges to children with rare or life-limiting chronic conditions and their families can be termed *psychosocial distress*. Pediatric psychosocial distress in this clinical context has been conceptualized in a variety of ways. Kazak et al. developed the widely used Pediatric Psychosocial Preventative Health Model (PPPHM), which employs a public health framework to match family psychosocial risk with appropriate interventions. The Psychosocial Assessment Tool (PAT) is a parent-reported screening tool based on the PPPHM that operationalizes psychosocial risk into the following domains: family structure/resources, family problems, social support, stress reactions, family beliefs, child problems, and sibling problems. Additional approaches to categorizing psychosocial risk among families with children who have rare or life-limiting chronic conditions include the Distress Thermometer, which screens for distress in domains related to practical, familial, emotional, and physical problems, as well as spiritual and religious concerns. Although no single definition exists for psychosocial distress among children with rare or life-limiting conditions, current approaches all take a broad, social-ecological approach that includes not only the psychological wellbeing of the patient but that of their caregivers and siblings, as well as their socio-economic circumstances, family structure, and social function. 

While it is clear that children with rare or life-limiting chronic conditions and their families are at increased risk for psychosocial distress, the extent to which this phenomenon and its antecedents are documented during routine medical appointments is unknown. Without adequate or standardized documentation of psychosocial concerns, families who would likely benefit from further assessment, targeted referrals, and service linkages may be at risk of slipping through the cracks. Therefore, the goal of the current study was to characterize the current medical record documentation practices around child and family psychosocial distress and risk factors for distress. Specifically, this work sought to answer the following research questions: (1) What information about patient and family psychosocial distress is extractable from the narrative text of clinical notes? (2) How frequently is psychosocial distress or its antecedents mentioned in routine outpatient clinical notes? and (3) How is psychosocial distress characterized by providers in the medical record? 

## 2. Materials and Methods

Eligible patients were children and adolescents (<20 years old) followed by a pediatric complex care coordination program at Mayo Clinic in Rochester, MN, USA (henceforth referred to as “the program”). The program is not a medical home but rather a consultative service that serves as the primary point of contact for patients and their families while receiving care from multiple specialty groups within Mayo Clinic. Approximately half of enrolled patients have at least one disease considered to be rare (i.e., affecting less than 200,000 Americans). The goal of the program is to improve communication among specialists at Mayo Clinic with local primary care providers and families to ensure a unified and holistic view of treatment plans and goals. Children who are followed by the program typically have significant chronic conditions in three or more body systems, need ongoing subspecialty care (longer than one year), and receive most of their subspecialty care at Mayo Clinic.

The program maintains a list of active patients (*n* = 166 at time of study), which was shared with the study team. Using a random number generator, 60 eligible patients from this list of active patients were identified for inclusion in our chart review. The most recent clinic visit with one of the program pediatricians or nurse practitioners was identified and the associated note was extracted for each patient in our sample. Two co-authors (EG and DG) each independently extracted five randomly selected patients to pilot test the extraction form and ensure consistency, which was deemed satisfactory after comparing responses and reaching consensus through discussion as a team. The remaining charts were divided between EG and DG and extracted individually. The following data elements were extracted from each patient’s chart using a REDCap [34] electronic data capture form: visit date and provider; demographic information, including patient age, gender, race, city and state of residence, preferred language, and health insurance; problem list; and the full narrative text of the note. Analysis proceeded according to a mixed methods explanatory sequential design (Figure 1).

### 2.1. Qualitative Analysis

Qualitative analysis was performed using NVivo qualitative data analysis software, Version 12. All clinical notes in the sample (*n* = 60) were coded on a line-by-line basis for narrative sections of the note (e.g., “History of Present Illness”, “Assessment”). The narrative text of the extracted encounter notes was coded using a mixed deductive (i.e., identified a priori based on existing literature on psychosocial distress in this population) and inductive (i.e., developing emergent codes that arose through review of the notes) approach. Key sources for deductive coding included domains from Kazak’s Psychosocial Assessment Tool (PAT) and the pediatric Distress Thermometer. An initial coding scheme was outlined based on previously described domains of psychosocial risk for children with serious medical illnesses and their families [35,36,37,38,39,40,41]. Authors then read through the narrative text of each note to further develop and refine the initial coding scheme, including determining which specific topics of discussion would be included in each broader coding category. After the development of the coding scheme, coding was performed on five notes in triplicate (SM, EG, and DG), using consensus to arrive at final code assignments for each note. Disagreements were taken as an indicator that the coding scheme required further clarification and the codebook was expanded and clarified as needed, eventually resulting in a final coding scheme organized into the following higher-order domains: Child Psychosocial Health, Family Adjustment and Support, and Family Structure and Resources. The final domains and subdomains reported on in this manuscript are detailed in Table 1 below. Each note was coded independently by two authors (DG, JC, EG). Any discrepancies between coding decisions were discussed as a team, with the third coder responsible for making final decisions in cases where disagreements could not be resolved. Quotes are used within the results section for illustrative purposes.

### 2.2. Quantitative Analysis

After the authors completed the coding of all 60 notes, descriptive statistics were generated for demographic information and visit diagnosis counts. Additionally, the number of notes in which each code appeared was counted. These counts acted as a proxy for the frequency in which discussion and documentation occurred around the respective topics in our sample of 60 patients. 

## 3. Results

### 3.1. Sample Characteristics

Demographic information is shown in Table 2. Patients included in this study had a mean age of 7.3 years (ranging from 10 months to 18 years). Slightly over half (56.7%) of patients were male. Most of the sample (91.7%) was white. All patients’ preferred language was listed as English. Twenty-five percent of patients had private insurance, 40.0% had Medicaid, and 35.0% percent had a combination of the two. Patients’ average drive time to receive care at Mayo Clinic was 187.6 min. The most common visit-related diagnosis categories were neurologic/neuromuscular (63.3% of visits), gastrointestinal (53.3%), and respiratory (41.7%). 

### 3.2. Child Psychosocial Health

Problems or concerns specifically regarding the patient’s emotional function were mentioned in 30 of the 60 medical notes (50%). While some notes described specific behaviors (e.g., “She does not listen to what her parents ask her to do”) others were vaguer (e.g., “[patient] has been struggling with social media interactions”). Difficulties with the patient’s behavior (*n* = 9) and anxiety (*n* = 4) were discussed most frequently. Treatment for these concerns was discussed in seven notes, usually by referencing ongoing treatment (e.g., “he is followed locally by a psychiatrist to manage his ADHD and anxiety and is scheduled to see a local counselor for his anxiety” and “she has benefited from her ABA program”) or the discussion of a referral for further evaluation and treatment (*n* = 1). The child’s school situation was mentioned in a majority of notes (*n* = 49; 81.7% of notes), with providers typically noting the patient’s current grade level at a minimum. However, many notes contained additional context on the child’s school function (e.g., “School is not going well due to fatigue”; “has attended 4 days of school this year due to Make A Wish, illness and appointments”). Concerns related to development were less common (*n* = 18; 30%) but still present in nearly one-third of notes in our sample. Finally, remarks about social interaction were coded in one-fifth of notes (*n* = 12; 20%) and included references to both positive experiences related to patient extracurricular activities, hobbies, and friendships, as well as difficulties related to participating in activities and socializing (e.g., “She is having a lot of urinary incontinence. This is making it difficult socially for her.”).

### 3.3. Family Adjustment and Support

Although 10 notes (16.7%) referred to parental emotional function, none of them identified specific concerns regarding a parent’s emotional health (e.g., parent mental health diagnosis; parent mental health treatment). Five notes referenced how the child’s medical illness and caregiving responsibilities were a source of significant stress on the parent (“[Parent] feels she is ‘hanging on by a thread”). Respite care was the only potential intervention mentioned in response to these stressors. It is interesting to note, some of the notes had a family history section which included parental mental health history; however, whether this section was included depended on the template the author used, and we were not able to ascertain who had entered these data into the medical record, or when it was entered. Therefore, this information was not included in our analyses. 

Seven of the 60 notes (11.7%) referenced the provision of social and material support to the parents or family. Examples of this included grandparents being trained to provide care for a patient, a patient’s mother who expressed feeling well-supported by resources being provided from the county, and another mother who was receiving extra supplies from a friend. A small number of notes (*n* = 5; 8.3%) mentioned instances of intra-family conflict (e.g., “Family has been under some stress as father is no longer involved with [child’s] care”). Finally, concerns regarding sibling emotional function were mentioned in 3 of the 60 notes (5%). In one case, a sibling’s diagnosis of autism was mentioned, and their aggression towards the patient was discussed as an ongoing stressor. In other cases, siblings were described as also having chronic health conditions, sometimes similar to those of the patient (e.g., “The family has been incredibly busy with managing three children (two with chronic health conditions)”.

### 3.4. Family Structure and Resources

Information about family structure was included in 47 of the 60 notes (78.3%). The most common descriptor was who the child lived with (*n* = 43), although specific information about risk factors, such as the age of these family members, was rarely included. Notes frequently mentioned the parents’ occupation (*n* = 29) if they were employed but did not include additional information about unemployment or underemployment. If the parents were not married, or were going through a divorce, this was also mentioned in some of the notes (*n* = 9). Nearly one-third of notes (*n* = 18) referenced barriers or facilitators to obtaining medical care and following treatment plans (e.g., “…as travelling to Rochester is a challenge for the family, they would like to limit medical appointments). However, financial concerns were specifically mentioned in only seven notes (11.7%). Most of these referenced struggles with obtaining insurance authorization or appealing denials for coverage of specific tests (e.g., genome sequencing), medications (e.g., injections), or services (e.g., increased nursing hours). Information regarding the impact of the child’s medical illness on parental employment (e.g., mother left the workforce to care for the child; stress taking care of the child and the family business simultaneously) were included explicitly in five notes (8.3%).

## 4. Discussion

In this chart review and analysis of pediatric outpatient encounter notes, we found that psychosocial distress and/or risk factors for distress were not consistently documented. Only one-third of notes in our sample documented discussion of the child’s emotional health, while mention of parent’s emotional concerns was largely absent—despite recent national data indicating that almost 20% of parents of children with rare or life-limiting chronic conditions report poor or fair mental health [23]. In addition, most notes in our sample did not capture risk factors and vulnerabilities of the broader family system (i.e., financial, parent and sibling adjustment). Although financial concerns were discussed in 12% of visits in our sample, survey data indicate much higher rates of financial hardship among this population [20], as well as high rates of unmet healthcare needs due to cost [42]. 

It is important to note that we do not intend for these findings to be critical of individual clinicians, as the reasons for lack of documentation are multifactorial and, in most cases, cultural and institutional. First, one key limitation of medical record data in the absence of validation methods (e.g., recording or observation of the clinical visit) is the inability to shed light on discussions that occurred but were not explicitly documented. Providers may be discussing psychosocial concerns to some degree during appointments but may feel reluctant to incorporate this information into medical notes for many reasons, including ambiguity about which problems rise to the level of clinical significance and which are a function of temporary stressors [43]. In addition, providers may also work under the assumption that psychosocial information is better covered in notes by social work or psychology. However, given the direct association between patient and caregiver psychosocial distress and medical outcomes, we argue that this information should also be included in medical provider documentation. Finally, psychosocial concerns in this population are often not limited to patients but are linked to family circumstances or parental factors. Providers may feel that questions about—let alone, formal documentation of—parental mental health or stressors are perceived as intrusive. However, the importance of including parents and caregivers in psychosocial assessments needs to be stressed, since these concerns are inextricably linked to the child’s well-being [44]. 

For both patients and their caregivers, universal psychosocial screening offers an opportunity to normalize the psychosocial impact of a child’s illness on the child and the family, and proactively identify children and families who may be experiencing current psychosocial distress or who are at risk for distress during the course of medical care [40,45,46]. Feedback about psychosocial concerns provided to clinicians through screening tools has been found to systematically increase discussion of emotional and psychosocial functioning [47]. Evidence suggests both pediatric clinicians and the parents of their patients support the practice of documenting psychosocial and mental health information in the patient’s health record [43]. 

There are several evidence-based tools available for screening for psychosocial risks and concerns in pediatric populations. These include the Distress Thermometer [39,40,48], the Psychosocial Assessment Tool 3.0 [37] and Checking In [49]. Research in pediatric oncology [37,39,49] (e.g., Kazak et al., 2018; Patel et al., 2019; Wiener 2021), organ transplant [50] and other life-threatening conditions [41] has demonstrated that psychosocial screening is feasible and acceptable to patients, caregivers, and medical providers. Systematic and routine psychosocial screening provides the opportunity to match the psychosocial care to the specific needs of the child and family, including providing further assessment, preventative interventions, and more specific evidenced-based care [46], with the goal of improving overall quality of life for the patient and their family [51]. The implementation of psychosocial screening increases the number of performed and accepted referrals to psychosocial providers [52], and vastly improves documentation of psychosocial concerns [53]. Screening tools also provide a starting point for clinicians, patients, and families to ease into what may be difficult or awkward conversations. In addition, several of these existing tools include instruments tailored for pediatric patients themselves to answer, allowing older children and adolescents an opportunity to actively participate in these discussions. However, further research is needed to inform best practices around implementation of pediatric psychosocial screening programs as well as the long-term impacts of screening on process and outcome metrics related to how effectively this information is integrated into clinical care [44,54,55]. It is important to note that psychosocial screening in the absence of appropriate referral or intervention strategies will not be sufficient to improve outcomes.

There are several limitations of our study. Our findings are a function of provider documentation in a single clinical note at one academic institution, which may limit the generalizability of these results. Additionally, the clinic notes assessed were from a complex care program which is not designated as a primary care medical home because the majority of patients do not live in the clinic’s immediate vicinity. Therefore, it is certainly possible that patients and families may be receiving social services in their local communities. However, even if psychosocial concerns are being discussed at a higher frequency than our study would imply, or if a patient and their family are receiving social support locally, we argue that documentation of these issues in the medical records of children with rare or life-limiting chronic conditions are critical to ensure surveillance, follow-up, and care coordination. 

Another limitation of our study is that data are from patients at a single academic medical center, limiting the generalizability of these results to other settings. In addition, only a single note from a medical provider at our institution was evaluated for this study, leaving the possibility that other notes could have mentioned topics related to psychosocial distress. However, we argue that these discussions should be occurring frequently, if not during each visit. Patients and their families may also be receiving social services outside our hospital system, especially given that many patients in the program do not live in the immediate vicinity of Mayo Clinic.

## 5. Conclusions

Children with rare or life-limiting chronic conditions and members of their families are at increased risk for the development of psychosocial distress, which, if left unidentified and untreated, can negatively impact the child and the family. The purpose of this research was to identify which elements of psychosocial distress and its antecedents are documented in the medical record, as well as their frequency and nature. While information related to family structure and patient school status was widely documented in our sample, many other important psychosocial domains (namely, child and family emotional function) were not routinely documented within medical provider notes. As these children continue to survive and live longer, they and their families may benefit from universal psychosocial screening, and an integrated medical and behavioral service model which could provide an evidence-based system of care encompassing more of the patient’s lived experience (Appendix A).

## Figures and Tables

**Figure 1 children-09-00664-f001:**
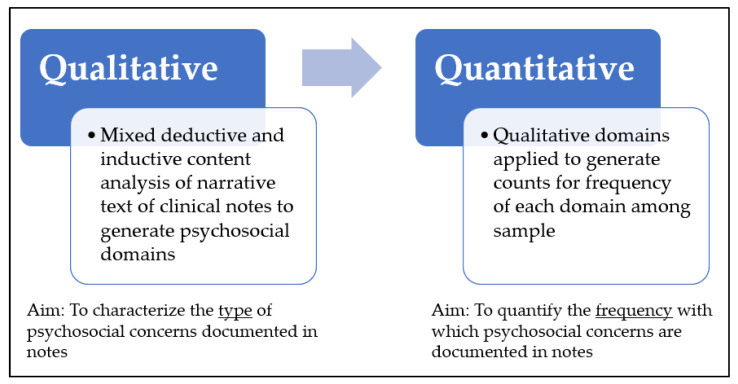
Mixed methods explanatory sequential design.

**Table 1 children-09-00664-t001:** Psychosocial domains developed during qualitative coding, their descriptions, and their prevalence in the study sample of clinical notes (*n* = 60).

Psychosocial Domains	Description	Frequency (%)
Child Psychosocial Health		
Developmental	Status of age-specific milestones (i.e., behavioral and/or physical skills associated with normal development).	18 (30.0)
Emotional function	References to child’s ability to regulate emotional expression and identify emotional expressions of others. Includes references to mood, behavioral, or attention problems, exposure to trauma, aggression, past or current therapy, and past or current psychiatric medication.	30 (50.0)
School	Mentions of school attendance, homebound, homeschool, grade; types of classes (e.g., special education); services received (e.g., PT, OT, speech therapy); Individualized Education Program (IEP) or 504 plan; academic performance or behavioral issues related to school.	49 (81.7)
Social Interaction	Opportunities for interaction with same-age peers; presence of friends, peer relationships, involvement in activities.	12 (20.0)
Family Adjustment and Support		
Intra-Family Conflict	Patient not getting along with family; parents not getting along; other members of the household not getting along; parents have conflicting ideas about parenting or conflict around medical decision making.	5 (8.3)
Parent Emotional Function	References to parent ability to cope; mood issues (e.g., worry, anxiety, depression, sadness); excessive substance use; avoidance, hypervigilance, a disabling parent health concern, current or past therapy.	10 (16.7)
Parent Social Support	References to social support and resources available to or used by parents (e.g., community resources, friends, family).	7 (11.7)
Sibling Emotional and Physical Function	Descriptions of anxiety or other mood concerns; disruptive behavior; current or past medical conditions; presence of sibling rivalry or conflict.	3 (5.0)
Family Structure and Resources		
Family Structure	Descriptions of individuals who live in the patient’s home (e.g., one parent only, grandparents, siblings) and/or the individuals involved in the patient’s care.	47 (78.3)
Financial Concerns	Issues related to money problems (e.g., trouble paying bills), obtaining transportation, maintaining adequate health insurance, parent’s ability to work, government assistance, and housing quality (e.g., evidence of overcrowding, frequent moves/evictions, or health hazards).	7 (11.7)
Parent Ability to Navigate Health System	References to parent’s ability to take time off to attend medical appointments, arrange child care, and follow through on medical treatment plan.	18 (30.0)
Parent Work-Family Conflict	References to issues with parent’s work situation, including having difficult hours/shifts, being under-employed, or interaction between caregivers’ work and subsequent stress or inability to care for child.	5 (8.3)

**Table 2 children-09-00664-t002:** Characteristics of the patient sample (*n* = 60).

Variable	*n* (%)	Mean (SD)
Age		7.3 (4.4)
Gender		
Male	34 (56.7)	
Female	26 (43.3)	
Race		
White	55 (91.7)	
Asian	3 (5.0)	
American Indian or Alaska Native	1 (1.5)	
Other	1 (1.5)	
Insurance		
Private	15 (25.0)	
Medicaid	24 (40.0)	
Private and Medicaid	21 (35.0)	
Driving minutes from home to Mayo Clinic		187.6 (154.6)
Visit diagnoses per patient (Median (Range))		4.0 (1–13)
Visit diagnosis categories		
Neurological or neuromuscular	38 (63.3)	
Gastrointestinal	32 (53.3)	
Respiratory	25 (41.7)	
Congenital or genetic	23 (38.3)	
ENT	18 (30.0)	
Endocrine	14 (23.3)	
Cardiovascular	11 (18.3)	
Orthopedic	10 (16.7)	
Psychiatric	9 (15.0)	
Renal or genitourinary	9 (15.0)	
Hematologic/immunologic	5 (8.3)	
Dermatology	2 (3.3)	
Ophthalmic	2 (3.3)	

## Data Availability

Not applicable.

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
