# Peer review of "Documentation of Psychosocial Distress and Its Antecedents in Children with Rare or Life-Limiting Chronic Conditions"

_children, 2022, doi:10.3390/children9050664_

Round 1
Reviewer 1 Report
There are only a few very minor amendments which I think would enhance the clarity of the study. It would be useful to develop the odd graph to highlight how the qualitative and quantitative elements link together. Please also see the file containing some minor amendments which would enhance the overall paper.
Author Response
Point 1: There are only a few very minor amendments which I think would enhance the clarity of the study. It would be useful to develop the odd graph to highlight how the qualitative and quantitative elements link together.
Response 1: Thank you for your feedback. We have added a figure (“Figure 1”) to illustrate how the qualitative and quantitative aspects of our approach worked together.

Reviewer 2 Report
Thank you for this very interesting article, dealing with issues that are too little studied in the field of rare diseases. I am convinced that the issue of psychosocial risk is essential in the adapted care of patients, but also of their families and caregivers.
The idea of extracting unstructured data from medical files is very good, because this way we can be closer to what was said during consultations, even if doctors do not note everything in the files of course.
In spite of these very interesting points, there are for me serious methodological problems in this work.
First of all, I have reservations about the methodology of the qualitative analysis.
It seems to me that there should be a section on how the analysis was done, first thematically with the key themes, the big ideas and how this information was classified, and then an analysis by sentence, by word, by block of words that corresponded to the research question. How were the sub-themes identified and how was the information collected from the medical records organized into these sub-themes?
There is no general description of the corpus, and the credibility of the data was not established according to the triangular strategies that use a combination of specialized teams to evaluate the results. The rigor of the analysis is here to their problem. It would have been interesting to have a summary table of the themes and sub-themes, with the number of times the concept is present per file and also the number of references for each theme.
It is difficult for us to have a global visualization of your results.
It would also have been interesting to have a table with Verbatim, reported in a literal way. All this is usual in qualitative studies.
Also, all the information collected in the charts comes from a single clinical note in a single institution, you mentioned this in your discussion but I think this is a serious limitation to generalizing your results.
Your work would benefit from more clarity on methodology.
Author Response
Point 1: In spite of these very interesting points, there are for me serious methodological problems in this work. First of all, I have reservations about the methodology of the qualitative analysis. It seems to me that there should be a section on how the analysis was done, first thematically with the key themes, the big ideas and how this information was classified, and then an analysis by sentence, by word, by block of words that corresponded to the research question. How were the sub-themes identified and how was the information collected from the medical records organized into these sub-themes?
Response 1: Thank you for raising this important point. We have updated Section 2.1 (“Qualitative analysis”) to include more detailed information about our qualitative methodology.
Point 2: There is no general description of the corpus, and the credibility of the data was not established according to the triangular strategies that use a combination of specialized teams to evaluate the results. The rigor of the analysis is here to their problem.
Response 2: We are unclear as to whether you are referring to the corpus of medical notes from which our data are drawn or the data compiled for analysis. If the former, we have noted in the Discussion that our data are subject to the same limitations inherent in all medical records. If the latter, we have attempted to describe in our Methods the steps we took to ensure the rigor of our analysis, including extraction and coding of notes in duplicate.
Point 3: It would have been interesting to have a summary table of the themes and sub-themes, with the number of times the concept is present per file and also the number of references for each theme. It is difficult for us to have a global visualization of your results.
Response 3: Please see Table 1 in the main text, which we have added in response to this point.
Point 4: It would also have been interesting to have a table with Verbatim, reported in a literal way. All this is usual in qualitative studies.
Response 4: Thank you for this point. We have included representative verbatim quotes throughout the narrative of the results sections.
Point 5: Also, all the information collected in the charts comes from a single clinical note in a single institution, you mentioned this in your discussion but I think this is a serious limitation to generalizing your results.
Response 5: Yes, we agree that this is a major limitation. However, we feel we have been transparent about this fact in both the Methods and Discussion of our paper.
Point 6: Your work would benefit from more clarity on methodology.
Response 6: Thank you for the thoughtful feedback provided in the above points related to reporting of our methodology. We hope the updates made to our paper have enhanced the clarity of our methods.

Reviewer 3 Report
The topic of the manuscript is very interesting, actual, interdisciplinary and in accordance with the journal’s topic of interest.
We would like to point out some suggestions that might be useful for authors, in order to improve the manuscript before an eventual publication:
- We consider that the first part of the manuscript - “Introduction” is well formulated in terms of contextualisation of the problem. However, it would be very important that the authors should include also a part in which they present the significant “literature review” on the topic of this manuscript. It could help not only to the improvement of the manuscript, but also it could help the readers to better understand the problem of psychosocial distress among children with rare or life-limiting chronic conditions (the theoretical framework - especially psychological and medical theoretical perspectives - is very, very important)
- In the second part “Materials and methods”, it would be very useful if the authors mention whether they have started from a(several0 research questions or hypotheses. If yes, it would be recommended to mention; also, in “Conclusions” (which, in fact, is missing) part they could point our how their research answered to these research questions or (if there were hypotheses) if their research confirmed or not the hypotheses.
- We recommend that the authors should add a final part “Conclusions”. It would give to the manuscript a more accurate and clear structure
Author Response
Point 1: We consider that the first part of the manuscript - “Introduction” is well formulated in terms of contextualisation of the problem. However, it would be very important that the authors should include also a part in which they present the significant “literature review” on the topic of this manuscript. It could help not only to the improvement of the manuscript, but also it could help the readers to better understand the problem of psychosocial distress among children with rare or life-limiting chronic conditions (the theoretical framework - especially psychological and medical theoretical perspectives - is very, very important)
Response 1: Thank you for this important suggestion. We have updated the Introduction to include a paragraph describing prominent frameworks for conceptualizing psycosocial distress in this population.
Point 2: In the second part “Materials and methods”, it would be very useful if the authors mention whether they have started from a(several0 research questions or hypotheses. If yes, it would be recommended to mention; also, in “Conclusions” (which, in fact, is missing) part they could point our how their research answered to these research questions or (if there were hypotheses) if their research confirmed or not the hypotheses.
Response 2: As this research was exploratory in nature, we did not start from an a priori hypothesis. To clarify our objectives, we have added specific research questions to the introduction (added text underlined): “Therefore, the goal of the current study was to characterize the current medical record documentation practices around child and family psychosocial distress and risk factors for distress. Specifically, this work sought to answer the following research questions: 1) What information about patient and family psychosocial distress is extractable from the narrative text of clinical notes? 2) How frequently is psychosocial distress or its antecedents mentioned in routine outpatient clinical notes? and 3) How is psychosocial distress characterized by providers in the medical record?” We have also referenced this in the newly added Conclusion section.
Point 3: We recommend that the authors should add a final part “Conclusions”. It would give to the manuscript a more accurate and clear structure
Response 3: Thank you for bringing this to our attention. We have added a “Conclusion” section after the Discussion.

Round 2
Reviewer 2 Report
thank you for the corrections.